# Long-Term Neuropsychological Outcomes Following Temporal Lobe Epilepsy Surgery: An Update of the Literature

**DOI:** 10.3390/healthcare9091156

**Published:** 2021-09-03

**Authors:** Ioanna Alexandratou, Panayiotis Patrikelis, Lambros Messinis, Athanasia Alexoudi, Anastasia Verentzioti, Maria Stefanatou, Grigorios Nasios, Vasileios Panagiotopoulos, Stylianos Gatzonis

**Affiliations:** 1Department of Neurology, Evangelismos Hospital, Ipsilantou 45-47, 10676 Athens, Greece; 2First Department of Neurosurgery, School of Medicine, National and Kapodistrian University of Athens, 10676 Athens, Greece; ppatrikelis@gmail.com (P.P.); alexoudath@yahoo.gr (A.A.); nverentzioti@gmail.com (A.V.); marwstef@gmail.com (M.S.); sgatzon@med.uoa.gr (S.G.); 3Lab of Cognitive Neuroscience, Department of Psychology, Aristotle University of Thessaloniki, 54124 Thessaloniki, Greece; lmessinis@psy.auth.gr; 4Department of Psychiatry, University of Patras Medical School, 26504 Patras, Greece; 5Department of Speech and Language Therapy, School of Health Sciences, University of Ioannina, 45500 Ioannina, Greece; grigoriosnasios@gmail.com; 6Department of Neurosurgery, University of Patras Medical School, 26500 Patras, Greece; panagiotopoulos2000@yahoo.com

**Keywords:** refractory temporal seizures, neurosurgery, cognitive outcome, memory, long-term follow-up

## Abstract

We present an update of the literature concerning long-term neuropsychological outcomes following surgery for refractory temporal lobe epilepsy (TLE). A thorough search was conducted through the PubMed and Medline electronic databases for studies investigating neuropsychological function in adult patients undergoing resective TLE surgery and followed for a mean/median > five years period. Two independent reviewers screened citations for eligibility and assessed relevant studies for the risk of bias. We found eleven studies fulfilling the above requirements. Cognitive function remained stable through long-term follow up despite immediate post-surgery decline; a negative relation between seizure control and memory impairment has emerged and a possible role of more selective surgery procedures is highlighted.

## 1. Introduction

Epilepsy surgery is nowadays an evidence-based treatment strategy for patients with drug-resistant epilepsy [1,2]. According to the International League Against Epilepsy (ILAE) definition, pharmacoresistant epilepsy is defined as the failure of a patient’s seizures to respond to at least two antiepileptic medications that are appropriately chosen and used for an adequate period [3]. With positive short-term surgery outcomes being definite [4,5], currently, epilepsy surgery centers focus on reporting long-term outcomes from cohort surgical studies implementing a variety of treatment techniques. Reputable reviews have undertaken the task to thoroughly assess post-operative seizure outcome [6,7]. However, it has been long acknowledged that seizure freedom is just one characteristic of surgery outcome. For patients to make truly informed decisions regarding their treatment, they need to know its effect on their ability to work, to study and to socialize. Highlighting the need for more reliable studies reviewing the non-seizure outcomes, neuropsychological data have also been evaluated and much has been documented on long-term postoperative cognitive outcomes [8,9] Although, an association with temporal lobe (TL) surgery and progressive memory decline has been suggested [10,11], while, currently, cognitive function is considered to remain stable one year after surgery [12]. Various factors such as chronological age, seizure recurrence, burden of medication and type of surgery have been linked to long-term postoperative outcomes [13,14]. Despite the encouraging results, epilepsy surgery is still underutilized [15]. Although the importance of early referral has been repeatedly emphasized [16], the delay between a refractory focal epilepsy onset and its surgery still remains of about 15–20 years [17,18]. Admittedly, there are steps to be taken with a view to one of the most frequent chronic and disabling disorders.

We conducted a review of the literature on long-term neuropsychological outcomes following TLE surgery. Our aim was to provide clinicians and researchers with a comprehensive summary reflecting a critical point of view of the current evidence.

## 2. Methods

### 2.1. Data Sources

We performed a comprehensive literature search on PubMed with a restriction to full-length English articles published till November 2020, as well as reviews, original articles and book chapters, and consulted experts about other studies. We used the following search terms in various combinations: “refractory temporal seizures”, “neurosurgery”, “cognitive outcome”, “memory” and “long-term follow-up”.

### 2.2. Study Selection and Classification

Two independent reviewers applied the following study inclusion criteria:
Reports of >20 patients with a medical history of drug-resistant temporal lobe epilepsy (TLE), undergoing resective surgery;Patients older than 16 years old;A mean/median >5 years post-surgery follow-up;Outcomes explored included long-term postoperative neuropsychological data and possible associated predictive factors (Figure 1).

## 3. Results

Eleven studies explored long-term neuropsychological outcomes in adult patients undergoing TL surgery according to the above searching criteria. One study evaluating intelligence consistently reported no worsening of performance following long-term postoperative follow up (Baxendale et al., 2012), while two others reported slight IQ improvement especially in patients achieving seizure freedom (Engman et al., 2006; Alpherts et al., 2004). All studies looked at the long-term memory outcomes, with five showing a greater memory decline following left than right TL resection (Helmstaedter et al., 2003; Rausch et al., 2003; Paglioli et al., 2004; Baxendale et al., 2012; Helmstaedter et al., 2018), and two older ones a progressive cognitive decline and lower memory scores in (Helmstaedter et al., 2003; Rausch et al., 2003). These findings have been challenged by later data, where cognitive function remained stable at one year following surgery, showing no evidence of accelerated memory decline (Engman et al., 2006; Alpherts et al., 2006; Andersson-Roswall et al., 2010; Baxendale et al., 2012; Salvato et al., 2016; Helmstaedter et al., 2018). Four other studies provided an account on which variables influenced postoperative memory improvement (Baxendale et al., 2012; Salvato et al., 2016; Mathon et al., 2017; Helmstaedter et al., 2018), while one study compared selective (anterior temporal lobectomy-ATL) with nonselective (selective amygdalohippocampectomy-SAH) surgery approaches with regard to neuropsychological outcomes, showing that risks of cognitive and/or verbal memory impairment were greater in patients with ATL than in those with SAH (Mathon et al., 2017). (Table 1)

## 4. Discussion

It is well established that epilepsy surgery is an excellent treatment option for achieving seizure control in patients suffering refractory TLE. However, risk for memory impairment remains to be considered as a serious post-surgical result. So far, long-term neuropsychological outcomes following TLE surgery have been reported by various prospective studies [19,20]. Given that few reports have focused on the long-term neuropsychological consequences of TLE surgery, we attempted to provide a review of the literature, investigating neuropsychological function of adult patients undergoing resective TLE surgery and followed for a mean/median > five years period.

In 2003, Helmstaedter and associates [10] reviewed cognitive and memory outcomes in 147 surgically- and 102 medically-treated TLE patients. They reported that surgery anticipated decline whether compared to the medically treated group, particularly when performed to the dominant hemisphere (usually the left), or whether seizures continued following surgery. In the same year, Rausch and colleagues [11] evaluated late postoperative cognitive changes in TLE patients undergoing standardized TL resection. Likewise, they reported a progressive cognitive decline continuing 13 years post-surgery, while left (L)TLE patients showed an accelerated memory decline.

In the following years, various longitudinal investigations presented inconsistent findings as to the continuing and accelerated pattern of postoperative memory decline in TLE. Testing for the presence of continuing postoperative verbal memory deficits in TLE patients during a six-year follow-up interval, Alpherts and collaborators [21] firstly provided evidence for a dynamic verbal memory decline up to two years following left temporal lobectomy, which then levels off. Later, similar long-term follow up studies confirmed such findings. Engman and collaborators [22] reported no signs of accelerated cognitive aging for most patients 10 years post-surgery. A longitudinal prospective study further supported the cognitive stability view, whereas the premise of an ongoing progressive verbal memory decline following TL resection was finally declined [23], since no association between seizure outcome and verbal memory course received confirmation.

Since post-surgical cognitive course in general and memory impairment in particular remained an open issue, many authors were willing to identify important determinants, such as postoperative seizure control and age of surgery. By studying the relationship between postoperative memory decline and seizure outcome for over a five-year follow-up period, Baxendale and associates [9] put forward that those who experienced more post-operative seizures presented verbal and visual memory changes, pinpointing to the role of poor seizure control in progressive memory impairment [9]. Similarly, a further risk for postoperative memory decline was poor seizures control [11]. During a five-year post-surgery follow-up, 50–60% of patients suffered some verbal and figural memory loss, with long-term memory gains being less common (15%) after TL surgery. The cumulative effect of seizures on memory was similarly highlighted over the next years by showing that apart from seizure control, shorter epilepsy duration, younger age, and antiepileptic drugs (AEDs) withdrawal would predict a better memory outcome [14]. Others [12] suggested that major losses appear in the early postoperative period, at one-year follow up, while a few patients decline further. Precisely, when seizure free, only 17% of those undergoing left and 10% of those undergoing right TL surgery showed verbal memory losses, as compared to 37% with left and 20% with right TL surgery who continued having seizures. In summary, as to post-surgery seizure outcome, recovery is more frequently observed than continuing decline.

The approach to surgery was another crucial factor studied extensively in the recent years. The rational underling elective surgery approaches is avoiding lesions following resective surgery to eloquent areas of the temporal neocortex, not directly involved in seizure generation. Mathon and collegues [14] compared three surgical approaches: anterior temporal lobectomy (ATL), transcortical selective amygdalohippocampectomy (SAH), and transsylvian SAH. They suggested that transcortical SAH tends to minimize cognitive deterioration after surgery, with the other two techniques having similar effects.

As to the optimal extent of surgical resection in TLE, no specific neurosurgical approach seems to outweigh the others in terms of seizures control [24]. A review suggested that in 76.2% of works there was evidence for a better cognitive outcome following elective surgery (e.g., SAH) as to the selective (S)ATL, while 23.8% of them did not find various neurosurgical procedures to differ [25]. Important as it may seem, more research is required to fully evaluate possible interactions between surgery approach and long-term (>five years) neuropsychological outcome.

### Models of Cognitive and Memory Prognosis Following Surgery

In the realm of TLE surgery, the concept of cognitive reserve has been applied in two different models of hippocampal functioning (i.e., functional reserve vs. hippocampal adequacy), in relation to the risk for memory impairment following temporal lobectomy (TLY). The functional reserve model claims that the size of memory loss is related to the spare capacity of the contralateral temporal lobe to support memory functions following resection of the abnormal (ipsilateral) one. IAT (Intracarotid Amobarbital Test) injections contralateral to the side of epileptogenesis typically produce memory impairment, whereas in the non-epileptic hemisphere memory function remains intact following injections to its epileptic counterpart [26,27,28,29,30]. A non-significant relationship has been recorded by some studies between the functional reserve of the contralateral-non-epileptic temporal lobe as assessed by the IAT and memory changes following TLY. There is rising evidence that the functional adequacy of the tissue to be resected determines the nature and extent of postoperative memory loss. The majority of patients with significantly intact memory before surgery were adversely affected following TLY [31,32,33]. Likely, studies on memory functioning performing IAT injections to the non-epileptic hemisphere showed that patients with a good pre-surgery memory performance were at much greater risk for memory loss than those who performed poor at baseline [34,35,36]. A weak point of the functional adequacy model; however, is that it does not predict mild material-specific memory deficits following TLY. Although the contralateral temporal lobe alone does not determine the probability of memory loss following TL, its functional contribution should not be ignored, especially if we consider ample clinical evidence documenting the devastating consequences for memory following bilateral hippocampal damage [36]. There is strong evidence of an inverse relation between the risk of postoperative memory impairments and the functional adequacy of the surgical temporal lobe, mostly seen with respect to verbal memory and left MTLE patients, rather than the functional reserve of the contralateral hemisphere [37].

Outcome studies in epilepsy surgery have identified several factors that have repeatedly been shown to be predictive of a poor prognosis, including the initial response to pharmacotherapy, the underlying etiology, and a patient’s history of seizure frequency [38]. From a neuropsychological point of view, one may suggest that restricting surgery to lesional and nonfunctional tissue should help to minimize the cognitive losses resulting from surgery. On the other hand, the functional adequacy of the to-be-resected brain tissue appears to be a major determinant of the cognitive outcome after surgery [39]. Stimulated by the ongoing discussion on the cognitive advantages of selective epilepsy surgery over extended standard resections in temporal lobe epilepsy, advances in MRI acquisitions, PET, SPECT, simultaneous EEG and functional MRI, and electrical and magnetic source imaging can be used to infer the localization of epileptic foci and assist in the design of intracranial EEG recording strategies [40]. Naturally, the outcome of epilepsy surgery will depend not only on the pre-surgery brain network but also on how the surgery (i.e., its location and extent) will affect the brain network [41]. Understanding how structural network abnormalities relate to seizure and cognitive outcomes after temporal lobe epilepsy (TLE) surgery can improve prediction of surgical outcomes [42]. The current standard for individualized prediction of surgical outcome primarily relies on clinical variables [43]. However, combining multivariate data and predicting post-surgery seizure freedom and cognitive outcome, is crucial to inform clinical management decisions.

## 5. Conclusions

While neuropsychological outcome studies of long-term follow-up remain scarce, progress has been made through the recent years, thus enabling clinicians reach into some safe conclusions for neurocognition after epilepsy surgery. Through our review of the literature, cognitive stability appears to be a still valid assumption receiving empirical support. It is also acknowledged that whenever seizures are controlled and medication reduced, recovery is more frequently observed than continuing decline. Elegantly implemented selective surgical procedures seem to limit cognitive side effects following surgery. In conclusion, the decision to proceed to surgery remains a highly individualized procedure requiring patient-tailored clinical and theory-based neuropsychological approaches. A continuing growth of evidence will help both physicians and patients with this important decision-making process. Finally, further data in cognitive reserve studies is warranted to contribute both long-term neuropsychological prognosis and rehabilitation following TLE surgery.

## Figures and Tables

**Figure 1 healthcare-09-01156-f001:**
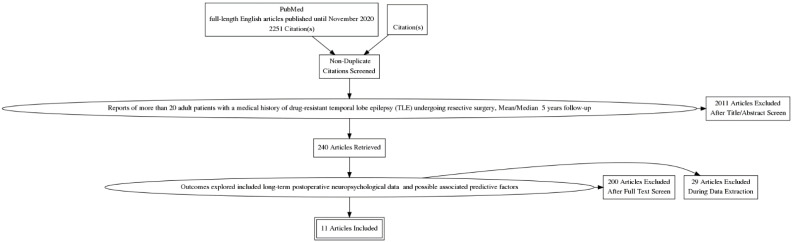
PRISMA flow diagram. The flow diagram depicts the flow of information through the different phases of a systematic review.

**Table 1 healthcare-09-01156-t001:** Neuropsychological outcome in studies with long-term follow up.

Scheme	Mean Follow-Up Years	Type of Surgery and *N* Sample Used	Population and Type of Study	Controlled Study	Neuropsychological Outcome
Helmstaedter, 2003	5	Temporal (*N* 147) Medical (*N* 102)	Adults prospective	Yes medical versus surgery	This was greater after a left temporal lobectomy or if seizures continued postoperatively. Seizure-free surgical patients showed a recovery of memory function. Intelligence: No significant changes were seen in either group
Rausch, 2003	12.8	Temporal (*N* 44) Medical (*N* 8)	Adults prospective	Yes medical versus surgery	Memory: Patients with LTL surgery showed selective early decreases in verbal memory. At the long-term follow-up, further decreases in verbal memory and visual memory scores were seen for all patient groups. The nonmemory scores remained stable over time.
Alpherts, 2004	6	Temporal (*N* 71)	Adults prospective	No	Intelligence: Right or left surgery did not affect intelligence
Paglioli, 2004	5.4	Temporal (*N* 65)	Adults prospective	No	Memory: Left side surgery: Of 38 patients, worsening occurred in logical memory in 5 (13%) and in verbal learning in 10 (26%). Right side surgery: Of 27 patients, worsening occurred in logical memory in one (4%), in verbal learning in three (11%), and in visual memory in 6 (22%).
Alpherts, 2006	6	Temporal (*N* 85)	Adults prospective	No	Memory: LTL patients showed an ongoing memory decline for consolidation and acquisition of verbal material for up to 2 years after surgery. RTL patients at first showed a gain in both memory acquisition and consolidation, which vanished in the long term. The group with pure MTS showed an overall lower verbal memory performance than the group without pure MTS (mesiotemporal sclerosis). A dynamic decline of verbal memory functions up to 2 years after left temporal lobectomy, which then levels off.
Engman, 2006	9.8	Temporal (*N* 25) Control group (*N* 25)	Adults prospective	Yes control group versus surgery	Memory: No signs of accelerated cognitive aging after 10 years in a majority of the patients. Those who were seizure-free at long-term follow-up had a significantly higher intelligence score than patients who were still having seizures
Andersson-Roswall, 2010	10	Temporal (*N* 51)	Adults prospective	No	Memory: Decline was detected already 2 years postoperatively, with no further decline from 2 to 10 years. The memory decline was not related to seizure outcome or AED treatment.
Baxendale, 2012	9.1	Temporal (*N* 71)	Adults prospective	No	Intelligence: No difference on intellectual function after surgery. Memory: Verbal learning LTL (Left temporal lobe) performed more poorly than the RTL (rlght temporal lobe). Visual learning: Patients who were seizure free at T4 demonstrated a significant improvement in visual learning. Patients who were not seizure free at the long term follow up had experienced a decline in visual learning. Those who were not stable both in verbal and visual memory had more post-operative seizures. Significant role of poor postoperative seizure control in progressive memory impairment suggesting cumulative effect of seizures on memory
Salvato, 2016	5	Temporal (*N* 151)	Adults retrospective	No	Memory: Patients with LTLE worsened in the immediate postsurgical period, their performance progressively improved, and at 5 years after surgery, it returned to be equal to the baseline. Shorter duration of epilepsy, younger age, and withdrawal of AED would predict a better memory outcome
Manton, 2017	8.7	Temporal (*N* 389)	Adults prospective	No	Worsening of cognitive function: Histology ILAE type 2/Preoperative verbal memory deficit/Surgical approach: ATL/Preoperative high seizure frequency/Advanced age at surgery/Surgery on left side/Postoperative major complications/Postoperative depression.
Helmstaedter, 2018		Temporal (*N* 161) Medical (*N* 208)	Adults retrospective	Yes surgery group versus medical group	Memory: In the operated group about 9% demonstrated significant losses in verbal memory, figural memory, or executive functions over the T2-T3 interval. In the nonoperated group 10%, 17%, and 6% showed a decline in verbal memory, figural memory, or executive functions between T1 and T3. 5–22 years after surgery, and compared to baseline, only 17% of those who had undergone left and 10% of those who had right temporal lobe surgery showed losses in verbal memory when they were seizure free, as compared to 37% of patients after left and 20% after right temporal lobe surgery if their seizures continued.

LTL = Left temporal lobe, RTL = Right temporal lobe, MTS = Mesial Temporal sclerosis, AED = antiepileptic drugs, LTLE = Left temporal lobe epilepsy, ATL = Anterior temporal lobectomy.

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
