# Peer review of "Long-Term Neuropsychological Outcomes Following Temporal Lobe Epilepsy Surgery: An Update of the Literature"

_healthcare, 2021, doi:10.3390/healthcare9091156_

Round 1

Reviewer 1 Report

This review is a good attempt and covered the topic reasonably well. Alexandratou and the group try to light on Cognitive function and TLE surgery and show a negative relation between seizure control and memory impairment. Seizure freedom post-surgery has been associated with the improved neurodevelopmental path. There are different rates of each type of surgical procedure and of presenting epilepsies amongst surgical candidates of varying age groups, which may also account for differences in outcome. The success of epilepsy surgery tends to be based on seizure outcome, i.e., change in severity and/or frequency of seizures after surgery and long-term postoperative memory outcome. This review is based on a comprehensive literature search; however, the authors have not discussed some important studies (Example enclosed), which can improve the conclusion and the review's impact.

For Example:

  • Grewe et al. 2019 https://doi.org/10.1016/j.seizure.2019.02.015
  • ArthurCukiert et al. 2019 https://doi.org/10.1016/j.seizure.2019.02.015
  • Leena Jutila et al. 2014 DOI: 1016/j.eplepsyres.2014.05.002
  • Leena Jutila et al. 2002 Journal of Neurology, Neurosurgery & Psychiatry 2002;73:486-494.
  • Katiuska Rosas et al. 2013 https://doi.org/10.1016/j.yebeh.2013.05.010
  • Jana Amlerova et al. 2013 https://doi.org/10.1016/j.yebeh.2012.10.025
  • Abuhuziefa Abubakr, Olukayode Onasanya 2012 doi: https://doi.org/10.4021/jnr149w

Author Response

Thank you for your kind review. We really appreciate you taking the time out to share your opinion. We aggree that some of the studies you attached, have not been mentioned. You are right about the importance of these studies. However, they were not included because they did not fit the profile of our selection criteria ( more than 20 patients, mean/median follow up>5years).

We indluded now two of the studies you mentioned ( Leena Jutila et al. 2002, 2004) as some really worth mentionig studies.

Thank you again for your review.

Reviewer 2 Report

Authors conducted a useful literature search regarding long-term neuropsychological outcomes following temporal lobe epilepsy surgery .They reported on 11 studies that fit their requirements for review. The stated conclusions as summarized, report stable cognitive function with follow-up, with a negative relation between seizure control and memory impairment. More selective surgery approaches were highlighted, discussing the concept of cognitive reserve. 

The authors should include recommendations for improved surgery selection based on the developmental origins and life-course theories, relevant to individuals who later present with intractable epilepsy. Earlier cerebral dysgenesis and destructive lesions span prenatal and postnatal ages that contribute to post-surgery cognitive/behavioral reserve as well as the success for seizure suppression. There is a continuity of risk from gene/environment interactions that begin from conception, through pregnancy, childhood into adulthood that cumulatively present multi-hit levels of risk.

Neuroinformatics combining relational clinical databases with high through-put genetic gain/loss of function biomarkers with volumetric/functional MRI studies will better reveal persons with aberrant connectivities over advancing ages. This approach ultimately will require  pediatric to adult data-sharing regarding epilepsy diagnosis and treatment protocols relevant to epilepsies and co-morbidities. Note the recent international classification (Blumcke et al 2021) that compares focal cortical dysplasia genetic panel analysis with histopathology to better detect subtypes. The currently reported studies in this review do not include this needed novel approach.

Author Response

Thank you for your review. We really appreciate you taking the time out to share your opinion. We aggree with the comments. Especially the last part, concerning the neuroinformatics and  function biomarkers.

A further research has been made and we added some comments in the end of discussion.

Thank you again for your time.